# Integer Discrete Flows and Lossless Compression

**Emiel Hoogeboom**[*]
UvA-Bosch Delta Lab
University of Amsterdam
Netherlands
e.hoogeboom@uva.nl

**Jorn W.T. Peters**[*]
UvA-Bosch Delta Lab
University of Amsterdam
Netherlands
j.w.t.peters@uva.nl

**Rianne van den Berg**[†]
University of Amsterdam
Netherlands
riannevdberg@gmail.com

**Max Welling**
UvA-Bosch Delta Lab
University of Amsterdam
Netherlands
m.welling@uva.nl

## Abstract

Lossless compression methods shorten the expected representation size of data without loss of information, using a statistical model. Flow-based models are attractive in this setting because they admit exact likelihood optimization, which is equivalent to minimizing the expected number of bits per message. However, conventional flows assume continuous data, which may lead to reconstruction errors when quantized for compression. For that reason, we introduce a flow-based generative model for ordinal discrete data called Integer Discrete Flow (IDF): a bijective integer map that can learn rich transformations on high-dimensional data. As building blocks for IDFs, we introduce a flexible transformation layer called integer discrete coupling. Our experiments show that IDFs are competitive with other flow-based generative models. Furthermore, we demonstrate that IDF based compression achieves state-of-the-art lossless compression rates on CIFAR10, ImageNet32, and ImageNet64. To the best of our knowledge, this is the first lossless compression method that uses invertible neural networks.

## 1 Introduction

Every day, 2500 petabytes of data are generated. Clearly, there is a need for compression to enable efficient transmission and storage of this data. Compression algorithms aim to decrease the size of representations by exploiting patterns and structure in data. In particular, *lossless* compression methods preserve information perfectly–which is essential in domains such as medical imaging, astronomy, photography, text and archiving. Lossless compression and likelihood maximization are inherently connected through Shannon's source coding theorem [34], i.e., the expected message length of an optimal entropy encoder is equal to the negative log-likelihood of the statistical model. In other words, maximizing the log-likelihood (of data) is equivalent to minimizing the expected number of bits required per message.

In practice, data is usually high-dimensional which introduces challenges when building statistical models for compression. In other words, designing the likelihood and optimizing it for high dimensional data is often difficult. Deep generative models permit learning these complicated statistical models from data and have demonstrated their effectiveness in image, video, and audio modeling

---

[*]Equal contribution
[†]Now at Google

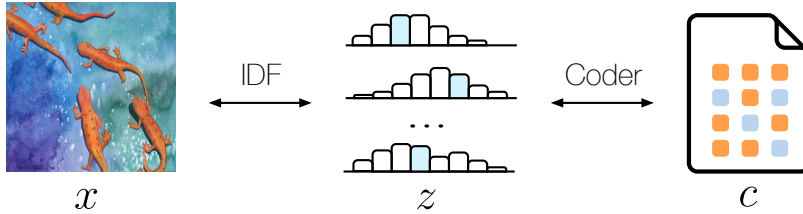

$$x \qquad\qquad z \qquad\qquad c$$

Figure 1: Overview of IDF based lossless compression. An image $x$ is transformed to a latent representation $z$ with a tractable distribution $p_Z(\cdot)$. An entropy encoder takes $z$ and $p_Z(\cdot)$ as input, and produces a bitstream $c$. To obtain $x$, the decoder uses $p_Z(\cdot)$ and $c$ to reconstruct $z$. Subsequently, $z$ is mapped to $x$ using the inverse of the IDF.

[22, 24, 29]. Flow-based generative models [7, 8, 27, 22, 14, 16] are advantageous over other generative models: *i)* they admit exact log-likelihood optimization in contrast with Variational AutoEncoders (VAEs) [21] and *ii)* drawing samples (and decoding) is comparable to inference in terms of computational cost, as opposed to PixelCNNs [41]. However, flow-based models are generally defined for continuous probability distributions, disregarding the fact that digital media is stored discretely–for example, pixels from 8-bit images have 256 distinct values. In order to utilize continuous flow models for compression, the latent space must be quantized. This produces reconstruction errors in image space, and is therefore not suited for lossless compression.

To circumvent the (de)quantization issues, we propose Integer Discrete Flows (IDFs), which are invertible transformations for ordinal discrete data–such as images, video and audio. We demonstrate the effectiveness of IDFs by attaining state-of-the-art lossless compression performance on CIFAR10, ImageNet32 and ImageNet64. For a graphical illustration of the coding steps, see Figure 1. In addition, we show that IDFs achieve generative modelling results competitive with other flow-based methods. The main contributions of this paper are summarized as follows: *1)* We introduce a generative flow for ordinal discrete data (Integer Discrete Flow), circumventing the problem of (de)quantization; *2)* As building blocks for IDFs, we introduce a flexible transformation layer called integer discrete coupling; *3)* We propose a neural network based compression method that leverages IDFs; and *4)* We empirically show that our image compression method allows for progressive decoding that maintains the global structure of the encoded image. Code to reproduce the experiments is available at https://github.com/jornpeters/integer_discrete_flows.

## 2 Background

The *continuous* change of variables formula lies at the foundation of flow-based generative models. It admits exact optimization of a (data) distribution using a simple distribution and a learnable bijective map. Let $f : \mathcal{X} \to \mathcal{Z}$ be a bijective map, and $p_Z(\cdot)$ a prior distribution on $\mathcal{Z}$. The model distribution $p_X(\cdot)$ can then be expressed as:

$$p_X(x) = p_Z(z) \left| \frac{dz}{dx} \right|, \quad \text{for } z = f(x). \tag{1}$$

That is, for a given observation $x$, the likelihood is given by $p_Z(\cdot)$ evaluated at $f(x)$, normalized by the Jacobian determinant. A composition of invertible functions, which can be viewed as a repeated application of the change of variables formula, is generally referred to as a normalizing flow in the deep learning literature [5, 37, 36, 30].

### 2.1 Flow Layers

The design of invertible transformations is integral to the construction of normalizing flows. In this section two important layers for flow-based generative modelling are discussed.

**Coupling layers** are tractable bijective mappings that are extremely flexible, when combined into a flow [8, 7]. Specifically, they have an analytical inverse, which is similar to a forward pass in terms of computational cost and the Jacobian determinant is easily computed, which makes coupling layers attractive for flow models. Given an input tensor $\mathbf{x} \in \mathbb{R}^d$, the input to a coupling layer is partitioned

into two sets such that $\mathbf{x} = [\mathbf{x}_a, \mathbf{x}_b]$. The transformation, denoted $f(\cdot)$, is then defined by:

$$\mathbf{z} = [\mathbf{z}_a, \mathbf{z}_b] = f(\mathbf{x}) = [\mathbf{x}_a, \mathbf{x}_b \odot \mathbf{s}(\mathbf{x}_a) + \mathbf{t}(\mathbf{x}_a)], \tag{2}$$

where $\odot$ denotes element-wise multiplication and $s$ and $t$ may be modelled using neural networks. Given this, the inverse is easily computed, i.e., $\mathbf{x}_a = \mathbf{z}_a$, and $\mathbf{x}_b = (\mathbf{z}_b - \mathbf{t}(\mathbf{x}_a)) \oslash \mathbf{s}(\mathbf{x}_a)$, where $\oslash$ denotes element-wise division. For $f(\cdot)$ to be invertible, $\mathbf{s}(\mathbf{x}_a)$ must not be zero, and is often constrained to have strictly positive values.

**Factor-out layers** allow for more efficient inference and hierarchical modelling. A general flow, following the change of variables formula, is described as a single map $\mathcal{X} \to \mathcal{Z}$. This implies that a $d$-dimensional vector is propagated throughout the whole flow model. Alternatively, a part of the dimensions can already be *factored-out* at regular intervals [8], such that the remainder of the flow network operates on lower dimensional data. We give an example for two levels ($L = 2$) although this principle can be applied to an arbitrary number of levels:

$$[\mathbf{z}_1, \mathbf{y}_1] = f_1(\mathbf{x}), \qquad \mathbf{z}_2 = f_2(\mathbf{y}_1), \qquad \mathbf{z} = [\mathbf{z}_1, \mathbf{z}_2], \tag{3}$$

where $\mathbf{x} \in \mathbb{R}^d$ and $\mathbf{y}_1, \mathbf{z}_1, \mathbf{z}_2 \in \mathbb{R}^{d/2}$. The likelihood of $\mathbf{x}$ is then given by:

$$p(\mathbf{x}) = p(\mathbf{z}_2) \left| \frac{\partial f_2(\mathbf{y}_1)}{\partial \mathbf{y}_1} \right| p(\mathbf{z}_1 | \mathbf{y}_1) \left| \frac{\partial f_1(\mathbf{x})}{\partial \mathbf{x}} \right|. \tag{4}$$

This approach has two clear advantages. First, it admits a factored model for $\mathbf{z}$, $p(\mathbf{z}) = p(\mathbf{z}_L)p(\mathbf{z}_{L-1}|\mathbf{z}_L)\cdots p(\mathbf{z}_1|\mathbf{z}_2, \ldots, \mathbf{z}_L)$, which allows for conditional dependence between parts of $\mathbf{z}$. This holds because the flow defines a bijective map between $\mathbf{y}_l$ and $[\mathbf{z}_{l+1}, \ldots, \mathbf{z}_L]$. Second, the lower dimensional flows are computationally more efficient.

## 2.2 Entropy Encoding

Lossless compression algorithms map every input to a unique output and are designed to make *probable* inputs *shorter* and *improbable* inputs *longer*. Shannon's source coding theorem [34] states that the optimal code length for a symbol $x$ is $-\log \mathcal{D}(x)$, and the minimum expected code length is lower-bounded by the entropy:

$$\mathbb{E}_{x \sim \mathcal{D}} \left[ |c(x)| \right] \approx \mathbb{E}_{x \sim \mathcal{D}} \left[ -\log p_X(x) \right] \geq \mathcal{H}(\mathcal{D}), \tag{5}$$

where $c(x)$ denotes the encoded message, which is chosen such that $|c(x)| \approx -\log p_X(x)$, $|\cdot|$ is length, $\mathcal{H}$ denotes entropy, $\mathcal{D}$ is the data distribution, and $p_X(\cdot)$ is the statistical model that is used by the encoder. Therefore, maximizing the model log-likelihood is equivalent to minimizing the expected number of bits required per message, when the encoder is optimal.

Stream coders encode sequences of random variables with different probability distributions. They have near-optimal performance, and they can meet the entropy-based lower bound of Shannon [32, 26]. In our experiments, the recently discovered and increasingly popular stream coder rANS [10] is used. It has gained popularity due to its computational and coding efficiency. See Appendix A.1 for an introduction to rANS.

## 3 Integer Discrete Flows

We introduce Integer Discrete Flows (IDFs): a bijective integer map that can represent rich transformations. IDFs can be used to learn the probability mass function on (high-dimensional) ordinal discrete data. Consider an integer-valued observation $x \in \mathcal{X} = \mathbb{Z}^d$, a prior distribution $p_Z(\cdot)$ with support on $\mathbb{Z}^d$, and a bijective map $f : \mathbb{Z}^d \to \mathbb{Z}^d$ defined by an IDF. The model distribution $p_X(\cdot)$ can then be expressed as:

$$p_X(x) = p_Z(z), \quad z = f(x). \tag{6}$$

Note that in contrast to Equation 1, there is no need for re-normalization using the Jacobian determinant. Deep IDFs are obtained by stacking multiple IDF layers $\{f_l\}_{l=1}^L$, which are guaranteed to be bijective if the individual maps $f_l$ are all bijective. For an individual map to be bijective, it must be one-to-one and onto. Consider the bijective map $f : \mathbb{Z} \to 2\mathbb{Z}$, $x \mapsto 2x$. Although, this map is a bijection, it requires us to keep track of the codomain of $f$, which is impracticable in the case of many dimensions and multiple layers. Instead, we design layers to be bijective maps from $\mathbb{Z}^d$ to $\mathbb{Z}^d$, which ensures that the composition of layers and its inverse is closed on $\mathbb{Z}^d$.

## 3.1 Integer Discrete Coupling

As a building block for IDFs, we introduce integer discrete coupling layers. These are invertible and the set $\mathbb{Z}^d$ is closed under their transformations. Let $[\mathbf{x}_a, \mathbf{x}_b] = \mathbf{x} \in \mathbb{Z}^d$ be an input of the layer. The output $\mathbf{z} = [\mathbf{z}_a, \mathbf{z}_b]$ is defined as a copy $\mathbf{z}_a = \mathbf{x}_a$, and a transformation $\mathbf{z}_b = \mathbf{x}_b + \lfloor \mathbf{t}(\mathbf{x}_a) \rceil$, where $\lfloor \cdot \rceil$ denotes a nearest rounding operation and $\mathbf{t}$ is a neural network (Figure 2).

Notice the multiplication operation in standard coupling is not used in integer discrete coupling, because it does not meet our requirement that the image of the transformations is equal to $\mathbb{Z}$. It may seem disadvantageous that our model only uses translation, also known as additive coupling, however, large-scale continuous flow models in the literature tend to use *additive* coupling instead of affine coupling [22].

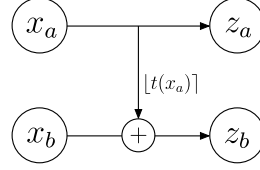

In contrast to existing coupling layers, the input is split in 75%–25% parts for $\mathbf{x}_a$ and $\mathbf{x}_b$, respectively. As a consequence, rounding is applied to fewer dimensions, which results in less gradient bias. In addition, the transformation is richer, because it is conditioned on more dimensions. Empirically this results in better performance.

Figure 2: Forward computation of an integer discrete coupling layer. The input is split in two parts. The output consists of a copy of the first part, and a conditional transformation of the second part. The inverse of the coupling layer is computed by inverting the conditional transformation.

**Backpropagation through Rounding Operation**
As shown in Figure 2, a coupling layer in IDF requires a rounding operation ($\lfloor \cdot \rceil$) on the predicted translation. Since the rounding operation is effectively a step function, its gradient is zero almost everywhere. As a consequence, the rounding operation is inherently incompatible with gradient based learning methods. In order to backpropagate through the rounding operations, we make use of the Straight Through Estimator (STE) [2]. In short, the STE *ignores* the rounding operation during back-propagation, which is equivalent to redefining the gradient of the rounding operation as follows:

$$\nabla_{\boldsymbol{x}} \lfloor \boldsymbol{x} \rceil \triangleq \boldsymbol{I}. \tag{7}$$

**Lower Triangular Coupling**
There exists a trade-off between the number of integer discrete coupling layers and the complexity of the layers in IDF architectures, due to the gradient bias that is introduced by the rounding operation (see section 4.1). We introduce a *multivariate* coupling transformation called Lower Triangular Coupling, which is specifically designed such that the number of rounding operations remains unchanged. For more details, see Appendix B.

## 3.2 Tractable Discrete distribution

As discussed in Section 2, a simple distribution $p_Z(\cdot)$ is posed on $\mathcal{Z}$ in flow-based models. In IDFs, the prior $p_Z(\cdot)$ is a factored discretized logistic distribution (DLogistic) [20, 33]. The discretized logistic captures the inductive bias that values close together are related, which is well-suited for ordinal data.

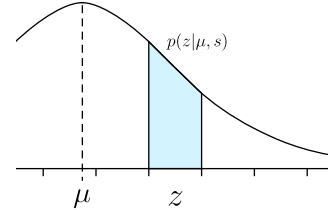

The probability mass $\text{DLogistic}(z|\mu, s)$ for an integer $z \in \mathbb{Z}$, mean $\mu$, and scale $s$ is defined as the density assigned to the interval $[z - \frac{1}{2}, z + \frac{1}{2}]$ by the probability density function of $\text{Logistic}(\mu, s)$ (see Figure 3). This can be efficiently computed by evaluating the cumulative distribution function twice:

Figure 3: The discretized logistic distribution. The shaded area shows the probability density.

$$\text{DLogistic}(z|\mu, s) = \int_{z - \frac{1}{2}}^{z + \frac{1}{2}} \text{Logistic}(z'|\mu, s)\mathrm{d}z' = \sigma\left(\frac{z + \frac{1}{2} - \mu}{s}\right) - \sigma\left(\frac{z - \frac{1}{2} - \mu}{s}\right), \tag{8}$$

where $\sigma(\cdot)$ denotes the sigmoid, the cumulative distribution function of a standard Logistic. In the context of a factor-out layer, the mean $\mu$ and scale $s$ are conditioned on the subset of

data that is *not* factored out. That is, the input to the $l$th factor-out layer is split into $\mathbf{z}_l$ and $\mathbf{y}_l$. The conditional distribution on $\mathbf{z}_{l,i}$ is then given as DLogistic($\mathbf{z}_{l,i}|\boldsymbol{\mu}(\mathbf{y}_l)_i, \mathbf{s}(\mathbf{y}_l)_i$), where $\boldsymbol{\mu}(\cdot)$ and $\mathbf{s}(\cdot)$ are parametrized as neural networks.

**Discrete Mixture distributions** The discretized logistic distribution is unimodal and therefore limited in complexity. With a marginal increase in computational cost, we increase the flexibility of the latent prior on $\mathbf{z}_L$ by extending it to a mixture of $K$ logistic distributions [33]:

$$p(z|\boldsymbol{\mu}, \mathbf{s}, \boldsymbol{\pi}) = \sum_k^K \pi_k \cdot p(z|\mu_k, s_k). \qquad (9)$$

Note that as $K \to \infty$, the mixture distribution can model arbitrary univariate discrete distributions. In practice, we find that a limited number of mixtures ($K = 5$) is usually sufficient for image density modelling tasks.

## 3.3 Lossless Source Compression

Lossless compression is an essential technique to limit the size of representations without destroying information. Methods for lossless compression require *i)* a statistical model of the source, and *ii)* a mapping from source symbols to bit streams.

IDFs are a natural statistical model for lossless compression of ordinal discrete data, such as images, video and audio. They are capable of modelling complicated high-dimensional distributions, and they provide error-free reconstructions when inverting latent representations. The mapping between symbols and bit streams may be provided by any entropy encoder. Specifically, stream coders can get arbitrarily close to the entropy regardless of the symbol distributions, because they encode entire sequences instead of a single symbol at a time.

In the case of compression using an IDF, the mapping $f : \mathbf{x} \mapsto \mathbf{z}$ is defined by the IDF. Subsequently, $\mathbf{z}$ is encoded under the distribution $p_Z(\mathbf{z})$ to a bitstream $\mathbf{c}$ using an entropy encoder. Note that, when using factor-out layers, $p_Z(\mathbf{z})$ is also defined using the IDF. Finally, in order to decode a bitstream $\mathbf{c}$, an entropy encoder uses $p_Z(\mathbf{z})$ to obtain $\mathbf{z}$. and the original image is obtained by using the map $f^{-1} : \mathbf{z} \mapsto \mathbf{x}$, i.e., the inverse IDF. See Figure 1 for a graphical depiction of this process.

In rare cases, the compressed file may be larger than the original. Therefore, following established practice in compression algorithms, we utilize an *escape bit*. That is, the encoder will decide whether to encode the message or save it in raw format and encode that decision into the first bit.

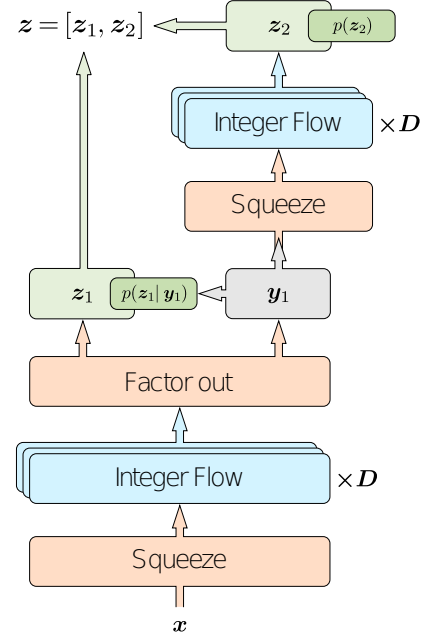

Figure 4: Example of a 2-level flow architecture. The squeeze layer reduces the spatial dimensions by two, and increases the number of channels by four. A single integer flow layer consists of a channel permutation and an integer discrete coupling layer. Each level consists of $D$ flow layers.

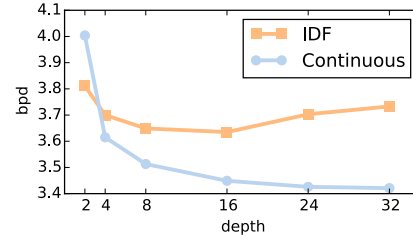

Figure 5: Performance of flow models for different depths (i.e. coupling layers per level). The networks in the coupling layers contain 3 convolution layers. Although performance increases with depth for continuous flows, this is not the case for discrete flows.

## 4 Architecture

The IDF architecture is split up into one or more levels. Each level consists of a squeeze operation [8], $D$ integer flow layers, and a factor-out layer. Hence, each level defines a mapping from $\mathbf{y}_{l-1}$ to $[\mathbf{z}_l, \mathbf{y}_l]$, except for the final level $L$, which defines a mapping $\mathbf{y}_{L-1} \mapsto \mathbf{z}_L$. Each of the $D$ integer flow layers per level consist of a permutation layer followed by an integer discrete coupling layer.

Following [8], the permutation layers are initialized once and kept fixed throughout training and evaluation. Figure 4 shows a graphical illustration of a two level IDF. The specific architecture details for each experiment are presented in Appendix D.1. In the remainder of this section, we discuss the trade-off between network depth and performance when rounding operations are used.

### 4.1 Flow Depth and Network Depth

The performance of IDFs depends on a trade-off between complexity and gradient bias, influenced by the number of rounding functions. Increasing the performance of standard normalizing flows is often achieved by increasing the depth, i.e., the number of flow-modules. However, for IDFs each flow-module results in additional rounding operations that introduce gradient bias. As a consequence, adding more flow layers hurts performance, after some point, as is depicted in Figure 5. We found that the limitation of using fewer coupling layers in an IDF can be negated by increasing the complexity of the neural networks part of the coupling and factor-out layers. That is, we use DenseNets [17] in order to predict the translation $\mathbf{t}$ in the integer discrete coupling layers and $\mu$ and $s$ in the factor-out layers.

## 5 Related Work

There exist several deep generative modelling frameworks. This work builds mainly upon flow-based generative models, described in [31, 7, 8]. In these works, invertible functions for continuous random variables are developed. However, quantizing a latent representation, and subsequently inverting back to image space may lead to reconstruction errors [6, 3, 4].

Other likelihood-based models such as PixelCNNs [41] utilize a decomposition of conditional probability distributions. However, this decomposition assumes an order on pixels which may not reflect the actual generative process. Furthermore, drawing samples (and decoding) is generally computationally expensive. VAEs [21] optimize a lower bound on the log likelihood instead of the exact likelihood. They are used for lossless compression with deterministic encoders [25] and through bits-back coding. However, the performance of this approach is bounded by the lower bound. Moreover, in bits back coding a single data example can be inefficient to compress, and the *extra bits* should be random, which is not the case in practice and may also lead to coding inefficiencies [38].

Non-likelihood based generative models tend to utilize Generative Adversarial Networks [13], and can generate high-quality images. However, since GANs do not optimize for likelihood, which is directly connected to the expected number of bits in a message, they are not suited for lossless compression.

In the lossless compression literature, numerous reversible integer to integer transforms have been proposed [1, 6, 3, 4]. Specifically, lossless JPEG2000 uses a reversible integer wavelet transform [11]. However, because these transformations are largely hand-designed, they are difficult to tune for real-world data, which may require complicated nonlinear transformations.

Around time of submission, unpublished concurrent work appeared [39] that explores discrete flows. The main differences between our method and this work are: *i)* we propose discrete flows for ordinal discrete data (e.g. audio, video, images), whereas they are are focused on categorical data. *ii)* we provide a connection with the source coding theorem, and present a compression algorithm. *iii)* We present results on more large-scale image datasets.

## 6 Experiments

To test the compression performance of IDFs, we compare with a number of established lossless compression methods: PNG [12]; JPEG2000 [11]; FLIF [35], a recent format that uses machine learning to build decision trees for efficient coding; and Bit-Swap [23], a VAE based lossless compression method. We show that IDFs outperform all these formats on CIFAR10, ImageNet32 and ImageNet64. In addition, we demonstrate that IDFs can be very easily tuned for specific domains, by compressing the ER + BCa histology dataset. For the exact treatment of datasets and optimization procedures, see Section D.4.

Table 1: Compression performance of IDFs on CIFAR10, ImageNet32 and ImageNet64 in bits per dimension, and compression rate (shown in parentheses). The Bit-Swap results are retrieved from [23]. The column marked IDF† denotes an IDF trained on ImageNet32 and evaluated on the other datasets.

| Dataset | IDF | IDF† | Bit-Swap | FLIF [35] | PNG | JPEG2000 |
|---|---|---|---|---|---|---|
| CIFAR10 | **3.34 (2.40×)** | 3.60 (2.22×) | 3.82 (2.09×) | 4.37 (1.83×) | 5.89 (1.36×) | 5.20 (1.54×) |
| ImageNet32 | **4.18 (1.91×)** | **4.18 (1.91×)** | 4.50 (1.78×) | 5.09 (1.57×) | 6.42 (1.25×) | 6.48 (1.23×) |
| ImageNet64 | **3.90 (2.05×)** | 3.94 (2.03 ×) | – | 4.55 (1.76×) | 5.74 (1.39×) | 5.10 (1.56×) |

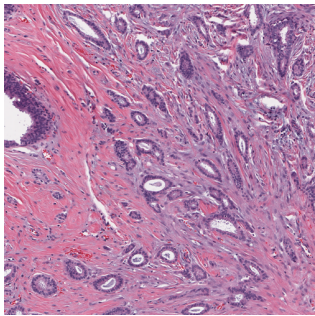 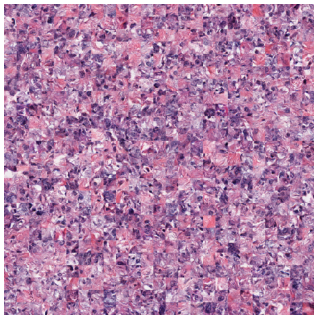 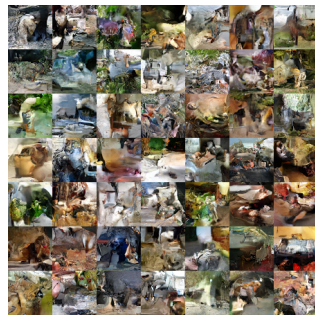

Figure 6: Left: An example from the ER + BCa histology dataset. Right: 625 IDF samples of size 80×80px.

Figure 7: 49 samples from the ImageNet 64×64 IDF.

## 6.1 Image Compression

The compression performance of IDFs is compared with competing methods on standard datasets, in bits per dimension and compression rate. The IDFs and Bit-Swap are trained on the train data, and compression performance of all methods is reported on the test data in Table 1. IDFs achieve state-of-the-art lossless compression performance on all datasets.

Even though one can argue that a compressor should be tuned for the source domain, the performance of IDFs is also examined on out-of-dataset examples, in order to evaluate compression generalization. We utilize the IDF trained on Imagenet32, and compress the CIFAR10 and ImageNet64 data. For the latter, a single image is split into four $32 \times 32$ patches. Surprisingly, the IDF trained on ImageNet32 (IDF†) still outperforms the competing methods showing only a slight decrease in compression performance on CIFAR10 and ImageNet64, compared to its source-trained counterpart.

As an alternative method for lossless compression, one could quantize the distribution $p_Z(\cdot)$ and the latent space $\mathcal{Z}$ of a continuous flow. This results in reconstruction errors that need to be stored in addition to the latent representation $\mathbf{z}$, such that the original data can be recovered perfectly. We show that this scheme is ineffective for lossless compression. Results are presented in Appendix C.

## 6.2 Tuneable Compression

Thus far, IDFs have been tested on standard machine learning datasets. In this section, IDFs are tested on a specific domain, medical images. In particular, the ER + BCa histology dataset [18] is used, which contains 141 regions of interest scanned at $40\times$, where each image is $2000 \times 2000$ pixels (see Figure 6, left). Since current hardware does not support training on such large images directly, the model is trained on random $80 \times 80$px patches. See Figure 6, right for samples from the model. Likewise, the compression is performed in a patch-based manner, i.e., each patch is compressed independently of all other patches. IDFs are again compared with FLIF and JPEG2000, and also with a modified version of JPEG2000 that has been optimized for virtual microscopy specifically, named JP2-WSI [15]. Although the IDF is at a disadvantage because it has to compress in patches, it considerably outperforms the established formats, as presented in Table 2.

Table 2: Compression performance on the ER + BCa histology dataset in bits per dimension and compression rate. JP2-WSI is a specialized format optimized for virtual microscopy.

| Dataset | IDF | JP2-WSI | FLIF [35] | JPEG2000 |
|---|---|---|---|---|
| Histology | **2.42 (3.19×)** | 3.04 (2.63×) | 4.00 (2.00×) | 4.26 (1.88×) |

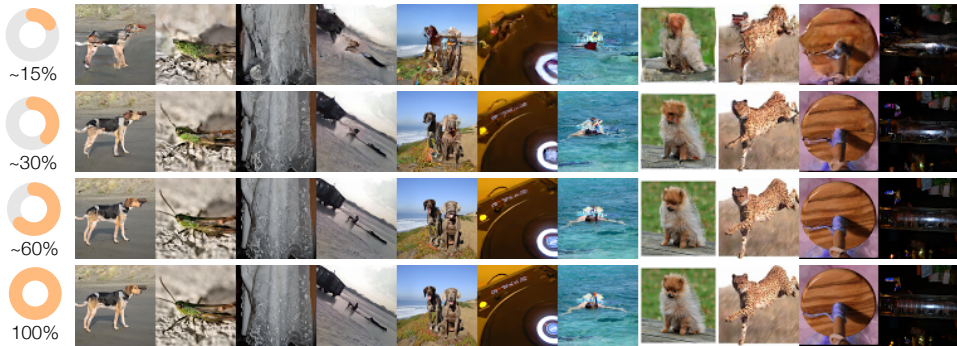

Figure 8: Progressive display of the data stream for images taken from the test set of ImageNet64. From top to bottom row, each image uses approximately 15%, 30%, 60% and 100% of the stream, where the remaining dimensions are sampled. Best viewed electronically.

### 6.3 Progressive Image Rendering

In general, transferring data may take time because of slow internet connections or disk I/O. For this reason, it is desired to progressively visualize data, i.e., to render the image with more detail as more data arrives. Several graphics formats support progressive loading. However, the encoded file size may increase by enabling this option, depending on the format [12], whereas IDFs support progressive rendering naturally. To partially render an image using IDFs, first the received variables are decoded. Next, using the hierarchical structure of the prior and ancestral sampling, the remaining dimensions are obtained. The progressive display of IDFs for ImageNet64 is presented in Figure 8, where the rows use approximately 15%, 30%, 60%, and 100% of the bitstream. The global structure is already captured by smaller fragments of the bitstream, even for fragments that contain only 15% of the stream.

### 6.4 Probability Mass Estimation

In addition to a statistical model for compression, IDFs can also be used for image generation and probability mass estimation. Samples are drawn from an ImageNet $32 \times 32$ IDF and presented in Figure 7. IDFs are compared with recent flow-based generative models, RealNVP [8], Glow [22], and Flow++ in analytical bits per dimension (negative $\log_2$-likelihood). To compare architectural changes, we modify the IDFs to *Continuous* models by dequantizing, disabling rounding, and using a continuous prior. The continuous versions of IDFs tend to perform slightly better, which may be caused by the gradient bias on the rounding operation. IDFs show competitive performance on CIFAR10, ImageNet32, and ImageNet64, as presented in Table 3. Note that in contrast with IDFs, RealNVP uses scale transformations, Glow has $1 \times 1$ convolutions and actnorm layers for stability, and Flow++ uses the aforementioned, and an additional flow for dequantization. Interestingly, IDFs have comparable performance even though the architecture is relatively simple.

Table 3: Generative modeling performance of IDFs and comparable flow-based methods in bits per dimension (negative $\log_2$-likelihood).

| Dataset | IDF | Continuous | RealNVP | Glow | Flow++ |
|---|---|---|---|---|---|
| CIFAR10 | 3.32 | 3.31 | 3.49 | 3.35 | 3.08 |
| ImageNet32 | 4.15 | 4.13 | 4.28 | 4.09 | 3.86 |
| ImageNet64 | 3.90 | 3.85 | 3.98 | 3.81 | 3.69 |

# 7 Conclusion

We have introduced Integer Discrete Flows, flows for ordinal discrete data that can be used for deep generative modelling and neural lossless compression. We show that IDFs are competitive with current flow-based models, and that we achieve state-of-the-art lossless compression performance on CIFAR10, ImageNet32 and ImageNet64. To the best of our knowledge, this is the first lossless compression method that uses invertible neural networks.

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
