[Supplementary Material]

# Integer Discrete Flows and Lossless Compression: Supplementary Material

**Emiel Hoogeboom**[*]
UvA-Bosch Delta Lab
University of Amsterdam
Netherlands
e.hoogeboom@uva.nl

**Jorn W.T. Peters**[*]
UvA-Bosch Delta Lab
University of Amsterdam
Netherlands
j.w.t.peters@uva.nl

**Rianne van den Berg**[†]
University of Amsterdam
Netherlands
riannevdberg@gmail.com

**Max Welling**
UvA-Bosch Delta Lab
University of Amsterdam
Netherlands
m.welling@uva.nl

## A  Additional background

### A.1  Asymmetric Numeral Systems

Asymmetric Numeral Systems (ANS) [1] is a recent approach to entropy coding. The range-based variant: rANS, is generally used as a faster replacement for arithmetic coding, because a state is only represented by a single number and fewer mathematical operations are required [2].

The encoding function of rANS encodes a symbol $s$ into a code $c'$ given the so far existing code $c$:

$$c'(c, s) = \lfloor c/l_s \rfloor \cdot m + (c \bmod l_s) + b_s, \tag{1}$$

where $m$ is a large integer that functions as the quantization denominator. Integers are chosen for $l_s$ such that $p(s) \approx {}^{l_s}\!/m$, where $p(s)$ denotes the probability of symbol $s$. Each symbol is associated with a unique interval $[b_s, b_s + l_s)$, where $b_s = \sum_{i=1}^{s-1} l_i$, as depicted in Figure 1.

Figure 1: The unique sequences for each symbol

The decoding function needs to retrieve the encoded symbol $s$, and the previous state $c$ from the new code $c'$. First consider the term $c' \bmod m$, which is equal to the last two terms of the encoding function: $c \bmod l_s + b_s$. This term is guaranteed to lie in the interval $[b_s, b_s + l_s)$. Therefore, the symbol can be retrieved by finding:

$$s(c') = t \ \text{ s.t. } b_t \leq c' \bmod m < b_{t+1}. \tag{2}$$

Consequently with the knowledge of $s$, the previous state $c$ can be obtained by computing:

$$c(c', s) = l_s \cdot \lfloor c'/m \rfloor + (c' \bmod m) - b_s. \tag{3}$$

In practice, $m$ is chosen as a power of two (for example $2^{32}$). As such, multiplication and division with $m$ reduces to bit shifts and modulo $m$ reduces to a binary masking operation.

---

[*]Equal contribution
[†]Now at Google

## B  Lower Triangular Coupling

There exists a trade-off between the number of integer discrete coupling layers and the complexity of the layers in IDF architectures, due to the gradient bias that is introduced by the rounding operation. For this reason, it is desired to increase the flexibility of layers without increasing the number of rounding operations. We introduce a *multivariate* coupling transformation called Lower Triangular Coupling, which is specifically designed such that the number of rounding operations remains unchanged. In practice, Lower Triangular Coupling does not offer significant improvements over standard coupling layers, and they both attain 4.15 bits per dimension (standard $\pm0.009$ and lower triangular $\pm0.007$), which is averaged over two runs with random weight initialization. The method is presented below for completeness.

The transformation of $\mathbf{x}_b$ is formed by multiplication with a strictly lower triangular matrix $\mathbf{L}$ which is conditioned on $\mathbf{x}_a$:

$$\mathbf{z}_b = \mathbf{x}_b + \lfloor \mathbf{t}(\mathbf{x}_a) + \mathbf{L}(\mathbf{x}_a)\mathbf{x}_b \rceil. \tag{4}$$

The main trick is to round the sum of all transformations, such that no additional gradient bias is introduced. This transformation is guaranteed to be invertible, and the inverse can be found with a modified version of forward substitution:

$$x_i^{(b)} = z_i^{(b)} - \left\lfloor t_i + \sum_{j=1}^{i-1} L_{ij} \cdot x_j^{(b)} \right\rceil, \tag{5}$$

where $x_i^{(b)}$ denotes the $i$th element of $\mathbf{x}_b$, and $\mathbf{t}$ and $\mathbf{L}$ are still conditioned on $\mathbf{x}_a$, however, this notation is dropped for clarity. The continuous case can even be solved analytically by using the inverse $\mathbf{x}_b = (\mathbf{I} + \mathbf{L})^{-1} (\mathbf{z}_b - \mathbf{t})$.

In practice we restrict the computational cost on feature maps $\mathbf{x}, \mathbf{z} \in \mathbb{Z}^{n_c \times h \times w}$ by parametrizing a *local* triangular matrix. That is, the transformation can be computed in parallel spatially, and is defined as: $\mathbf{z}_{:,vu}^{(b)} = \mathbf{x}_{:,vu}^{(b)} + \left\lfloor \mathbf{t}_{:,vu} + \mathbf{L}_{vu}\mathbf{x}_{:,vu}^{(b)} \right\rceil \forall vu$, where $v, u$ denote spatial coordinates, $\mathbf{L}_{vu} \in \mathbb{R}^{c_b \times c_b}$ and $\mathbf{t}$ are conditioned on $\mathbf{x}^{(a)}$, and $c_b$ denotes the number of channels in $\mathbf{x}^{(b)}$. Since the dimensions of $\mathbf{L}_{vu}$ are small, relative to the neural networks parametrizing them, the inverse can be found in $c_b$ iterations using spatially parallelized matrix operations.

## C  Quantizing a Continuous Flow

To test the lossless compression performance of continuous flows, the latent space is quantized to a linear spaced bins. Because the latent space is quantized, the reconstructions may contain errors. To enable lossless compression, FLIF is used to encode the errors in reconstruction. Hence, given the quantized latent variables and the reconstruction errors, the original input can be obtained.

Figure 2: Compression performance of a quantized continuous flow model using different bin sizes. The dashed line denotes the analytical bpd of the continuous model. The total required bpd consists of both the quantized latent $\mathbf{z}$ and the residual errors are encoded separately using the FLIF format.

The performance of the quantized flow is shown in Figure 2. When the bin size is large ($\frac{1}{128}$), encoding the latent representation requires relatively few bits, because the probability area is larger. However, the residuals are higher, and require more bits to be modelled. Analogously, when the bin size is small ($\frac{1}{512}$), encoding the latent representation requires more bits, but the residual can be modelled using fewer bits. Although the bits required for the residual or the quantized latents may be small individually, their sum is always large. In total the quantized flow performs poorly on lossless compression.

# D Experimental details

## D.1 Networks

The coupling and factor out layers are parametrized using neural networks. These networks are DenseNets [3]. Specifically we use $n = 512$ intermediate channels and a depth $d = 12$. In contrast with standard DenseNets, we do not use normalization layers. A single layer in the densenet consists of:

$$\text{Conv}1 \times 1 \rightarrow \text{ReLU} \rightarrow \text{Conv}3 \times 3 \rightarrow \text{ReLU},$$

## D.2 IDF architecture

The exact architecture for experiments is specified in Table 1. All models are trained using Adamax [5] with standard parameters. Furthermore, the learning rate is computed as: $lr = lr_{base} \cdot decay^{epoch}$. We follow the preprocessing procedure for CIFAR10 as described in [6]. For ImageNet32 and ImageNet64, we do use additional preprocessing. For the ER + BCa dataset, we employ random horizontal and vertical flips during training.

Table 1: IDF architecture and optimization parameters for each experiment.

| Dataset | $L$ | $D$ | densenet depth | densenet channels | batchsize | patchsize | train examples | lr decay | epochs |
|---|---|---|---|---|---|---|---|---|---|
| CIFAR10 | 3 | 8 | 12 | 512 | 256 | 32 | 40000 | 0.999 | 2000 |
| ImageNet32 | 3 | 8 | 12 | 512 | 256 | 32 | 1230000 | 0.99 | 100 |
| ImageNet64 | 4 | 8 | 12 | 512 | 64 | 64 | 1230000 | 0.99 | 20 |
| ER + BCa | 4 | 8 | 12 | 512 | 50 | 80 | 114 | 0.99999 | 50000 |

In our implementation, instead of using integers in $\mathbb{Z}$, we use the equivalent representation $\mathbb{Z}/256$, which we found to work better with standard weight initialization and optimization methods. Despite the fact that this implementation does not use integers, it is functionally equivalent to the method presented in the main text.

## D.3 Dataset preparation

The dataset for CIFAR10 originally consists of 50000 train images and 10000 test images. We use the last 10000 images for validation which results in 40000 train, 10000 validation and 10000 test images. ImageNet32 and ImageNet64 originally contain approximately 1250000 train and 50000 validation images. The validation images are used solely for testing, and 20000 images are randomly selected as a new validation set. This results in roughly 1230000 train, 20000 validation and 50000 test images.

The ER + BCa dataset [4] [3] is split into $114$ train images and $28$ test images such that specific patients IDs only occur in one of the two sets. The test patient identifiers are:

```
8915   8959   9023   9081   9256   9382   10264  10301
12749  16532  12818  12871  12884  12908  12931  12949
13106  13459  13459  13617  13694  14154  14305  16661
17117  17643  25289  25617
```

## D.4 Hardware and Software

The code for our experiments is implemented using PyTorch [7]. The model implementations are based on the codebase released along with [9] whereas the rANS coder implementation was taken from [8]. All experiments were run using 4 Nvidia GTX 1080Ti GPUs.