[Reviews · NeurIPS 2019]

Reviewer 1



(Originality): This is the first paper to discuss flow-based models for discrete data, concurrent with [Tran et al. 2019] (see https://arxiv.org/abs/1905.10347). However, there is little overlap between the papers as the other paper considers flows for nominal data and proposes a different set of flows. The paper also, to the best of my knowledge, is the first to use flow-based methods for lossless compression, concurrent with [Ho et al. 2019] (see https://arxiv.org/abs/1905.08500). (Quality): The paper appears technically sound. As the flows map from integers to integers, the straight-through estimator (STE) is used to allow gradient-based optimization. The authors are clear about the gradient bias introduced by the STE. (Clarity): I found the paper to be very readable and with a good structure. Background material is introduced, the method is clearly explained and supported by several illustrative figures. (Significance): This paper is among the first papers on two emerging lines of research: - Flows for discrete data. - Flows for lossless compression.

Reviewer 2



The paper proposes a discrete flow model (IDF) based on the ideas from NICE/Real NVP. It differs from another discrete flow model of Tran et al., though the differences in layers construction are not too big. I don’t know what the policy regarding similar ideas is, but this work has strong experiments achieving state-of-the-art results, it seems to offer a better design for the factor-out layers and the latent distribution, and the paper is well organised and clearly written. I think that (discrete) flow model is an important direction to work on and this paper will be used by others. I think the paper could be much better if it gave intuition on why some choices were made. For example, there is a conditional dependence between the parts of z. How much worse would the model be with independent latents? Another question is how much bigger does the model need to be if only translations are used instead of modulo scaling and translation? Have you tried using a softmax instead of a discretised logistic distribution? The results in Table 3 are surprising. It’s related to the questions above. Do you know why a simpler model with only translations performs better than Real NVP with both scaling and translation? Would the difference be due to the dependencies in z or a different latent distribution? Are the models in Table 3 comparable in terms of the number of parameters? Section 2.2 says that rANS entropy encoder was used in the experiments, so I would assume that compression results in Table 1 are computed after you applied rANS. Is that correct? So have you implemented an encoder-decoder pair that one can readily use as a replacement to lossless JPEG2000 or PNG? How would they compare with respect to computational costs? Line 244 typo: ‘they are are focused’. ========== UPDATE ============ The rebuttal was sufficient and I'm happily increasing my score.

Reviewer 3



The core contribution of this paper is the discrete coupling layer for discrete variables. This layer can be seen as a variant of the continuous coupling layer in Real NVP or Glow. That is, they are designed in a similar way that splitting the input into two parts, and using the first part to generate parameters for the second part. The main difference is that the discrete coupling layer uses the rounding operation to round the bias. The paper designs a discrete flow model based on the proposed discrete coupling layer. The model uses DLogistic as the distribution for latent code. The paper uses the proposed discrete normalizing flow to do lossless compression, and outperforms the current methods. Pros: 1. The paper is well written and easy to follow. 2. The proposed discrete coupling layer is useful for discrete variables. 3. The application is new, since the previous flow models are all applied to generate synthetic images. 4. The experiments are good and prove the ability of the model to do lossless compression. Cons: 1. I don’t quite understand why we should split the input to 75-25 parts. It seems that it is just arbitrarily set, so I think it needs to be discussed more, or some empirical results to prove it. 2. I think a better way to compare the proposed model with other flow models, e.g. Glow, is to compare the generated samples, and use larger images, e.g., 64x64, 96x96 or 128x128. The images in Figure 7 are so small that cannot prove the model’s ability of image generation. With only the NLL on small datasets, it is hard to say that the discrete flow performs as well as the state-of-the-art continuous flows, e.g., Flow++ and Glow. 3. Maybe I missed something. To generalize the discrete flow to continuous flow, I think we need to compute the determinant of the Jacobian matrix, but I did not see the authors mention it. ================================= I have read the authors' responses and the other reviewers' comments. I did not change my score. The reasons are as below. 1. The authors did not answer my question about why we need to split the variable to 75-25. 2. The authors did not provide an example of 64x64 generated image in the author responses. So I am not quite sure how good the generated images can be. 3. Overall, it is a good paper, and I tend to accept it.

[Author Response · NeurIPS 2019]

We thank the reviewers for their positive feedback and constructive comments. We will summarize and respond to the
comments of all the reviewers. All reviewers recognize our contributions as 1) being among the first to design flows for
discrete data, 2) building a bridge between generative flow models and compression. The reviewers see the topic of
discrete flows as emerging and important. Moreover, the reviewers agree that the paper is easy to read, clearly organized
and that the experiments are well chosen and demonstrate the compression capability of the proposed model. Reviewer
2 thinks that our paper will be used by others.

Reviewer 1 and 2 refer to concurrent work on discrete flows by Tran et al., which is acknowledged in both our paper
and theirs. Although there are clear similarities between the two approaches, i.e. both works use the straight-through
estimator, we focus on the use of discrete flows for ordinal data and explore connections between generative modelling
and data compression, whereas Tran et al. focus on nominal data and perform experiments in an NLP setting.

Reviewer 2 indicates that she/he is unsure if we have implemented an actual encoder-decoder pair. We want to emphasize
that this is in fact the case. The encoder is defined by the forward pass of the IDF flow model and the rANS encoder,
whereas the decoder consists of the rANS decoder and the inverse of the IDF. Hence, the combination of our IDF and
the rANS encoder are a direct replacement for conventional methods such as JPEG2000 or PNG. In addition, the results
presented in Table 2 are the *actual* compression rates obtained using the IDF + rANS encoder-decoder, given both as
bits per dimension (bpd) in the bitstream and compression rate.

Reviewer 3 is of the opinion that it is preferable to compare the proposed model to existing flow methods in terms of
their generative performance. It is argued that we should compare samples from our model at higher resolutions (e.g.
64x64 or 96x96 etc), and that the negative log likelihood (NLL) results are only reported on low-resolution datasets. We
would like to emphasize that table 3 includes NLL results on the 64x64 version of ImageNet. Hence we disagree with
the statement that we do not consider datasets with large enough images to be able to judge the generative modeling
performance from the NLL scores. However, we agree that the samples shown in figure 7 of our model trained on
ImageNet 32x32 are depicted too small. We will fix this by showing a smaller number of samples of an IDF model that
was trained on ImageNet 64x64.
Last but foremost, we stress that our main goal is to design a flow-based lossless compression algorithm for discrete
ordinal data such as images. We have demonstrated that our results are state-of-the art on this task in table 1 and 2. We
consider the task of generative modeling (results in table 3) as complementary.

Reviewer 2 would like to see a more elaborate discussion of the design choices in our model, such as the dependency
structure in the prior and the use of a discretized logistic prior. An intuitive argument for the dependency structure in
the prior is that it provides a natural ordering for the partial loading of a data stream as depicted in Figure 8. Note that
this type of structured prior is also used in Glow. A discretized logistic is preferred over a softmax prior because 1) a
softmax prior does not model the ordinality among discrete variables, as opposed to the discretized logistic; 2) IDF
can output any integer when mapping an image to latent space, even integers <0 and >255. It is therefore not trivial
to implement a categorical distribution using a softmax for this type of support. In contrast, a discretized logistic has
infinite support by design making it a more natural choice. 3) A softmax prior is more expensive in terms of the number
of parameters than a (mixture) of discretized logistic distributions. Each logistic distribution is determined by only 2
parameters. The number of required mixture components is not high either (we use only 5 in our paper).

As for the differences in the models in table 3 (mentioned by reviewer 2): we took Glow as a starting point for the
architecture design, and replaced the ResNets in the coupling layers with DenseNets. For ImageNet32, RealNVP uses
less parameters than IDF and its continuous version (CF) (46.2M[1] vs 58.5M), whereas IDF/CF is more efficient in the
case of ImageNet64 (84.3M vs 184.9M[1]). To reduce the effect of the bias from the straight through estimator for the
rounding operator we use fewer but more complex coupling layers as compared to Glow (See discussion section 4.1).
kThe second difference lies in the use of additive (translation) coupling layers, as opposed to affine (translation + scale
layers). However, we'd like to point out that Kingma & Dhariwal (2018) experimentally showed that the difference
between affine and additive coupling layers is fairly small. We have purposely included results for the continuous
version of IDF, such that comparing it with IDF on the one hand and RealNVP/Glow/Flow++ on the other hand allows
to distinguish the influence of discretization and architecture design. We hope this sufficiently addresses reviewer 2's
questions on the comparison between models in table 3.

Finally, reviewer 3 questions how the determinant of the Jacobian matrix is computed for the continuous generalization
of our model. The continuous version is obtained from IDF by removing the rounding operator. It therefore still only
contains translation/additive couplings, leading to a Jacobian determinant of one.

Again, we thank the reviewers for their positive and constructive feedback. We hope that we have sufficiently addressed
the remaining questions/comments of the reviewers, and we will incorporate the above clarifications into the final
version of our paper.

## Footnotes

[1]This number was obtained from the implementation released by Google


[Meta-Review · NeurIPS 2019]

This paper ports flow-based models to discrete data, concurrent with [Tran et al. 2019](https://arxiv.org/abs/1905.10347). It's technically sound, up-front about limitations, and has sensible experiments.